# Male Layer Chicken’s Response to Dietary *Moringa oleifera* Meal in a Tropical Climate

**DOI:** 10.3390/ani12141843

**Published:** 2022-07-20

**Authors:** Esther Faustin Evaris, Luis Sarmiento Franco, Carlos Sandoval Castro, Jose Segura Correa, Jesús Arturo Caamal Maldonado

**Affiliations:** Department of Animal Nutrition, Faculty of Veterinary Medicine and Animal Science, University of Yucatan (UADY), Km 15.5 Carretera Mérida-Xmatkuil, Apdo. 4-116, Itzimna, Merida 97100, Mexico; esther_faustin@yahoo.ca (E.F.E.); carlos.sandoval@correo.uady.mx (C.S.C.); jose.segura@correo.uady.mx (J.S.C.)

**Keywords:** finisher slow-growing, Dominant CZ Blue D 107, productive performance, outdoor access, carcass quality, blood parameters, moringa

## Abstract

**Simple Summary:**

In the egg industry, most slow-growing chicks are usually euthanized at 1-day-old even though it has been demonstrated that they are well-suited to production systems with outdoor access. However, the rearing of these birds, as an option to address the ethical concerns related to their disposal at such an early age, is impaired by high feeding costs. Therefore, it is crucial to evaluate non-conventional feeding materials that could be incorporated into their diets. Moringa is a non-traditional feed for poultry; nevertheless, studies investigating slow-growing breeds’ performance when fed Moringa while given access to outdoors in the tropics are limited. Thus, the current study evaluated slow-growing chickens’ response to dietary Moringa in terms of production performance, carcass yield, and blood parameters. The results revealed that all cumulative traits were not negatively affected by the treatments (except for feed conversion). Additionally, both breast weight and yield and gizzard weight and yield significantly increased with dietary Moringa, while blood parameters remained unaltered. It is concluded that Moringa, as a feed ingredient, could be added to the diets of slow-growing male chickens grown with outdoor access in the tropics to improve productive and carcass traits, but feed conversion could be increased.

**Abstract:**

This study was conducted to evaluate the effects of different dietary amounts of *Moringa oleifera* meal (MOM), consisting of leaves and small edible green stems, on growth performance, carcass traits, and blood parameters of finisher male layer-type slow-growing chickens raised with outdoor access. A total of 198 Dominant Blue D 107, 72-day-old male chickens were randomly assigned to tree dietary treatments containing 0, 30, and 60 g/kg MOM that corresponded to T1 or control, T2, and T3, separately. Each treatment was divided into 6 replicates of 11 birds apiece, and all birds had access to the outdoors. After 7 weeks of experimental rearing, live weight was 2218.1, 2164.4, and 2176.6 g for birds raised in T1, T2 and T3, respectively, without statistical differences (*p* ≥ 0.05). Live weight gain and daily live weight gain increased (*p* ≤ 0.05) in favor of the control group during the first 4 weeks but decreased (*p* ≤ 0.05) in the last 3 weeks of the trial. Overall feed intake was not influenced by the treatments (*p* ≥ 0.05). Birds raised with MOM in their diets had higher feed conversion (*p* ≤ 0.05). T2 diet considerably increased (*p* ≤ 0.05) birds’ breast and gizzard weight and yield while decreasing feet weight and yield, in comparison to T1. Dietary MOM inclusion did not impair or improve (*p* ≥ 0.05) blood parameters. The results of this study suggest that up to 60 g/kg MOM could be included in the diets of slow-growing Dominant Blue D 107 male chickens grown with access to outdoors without compromising their productive performance (except for feed conversion), slaughter characteristics, and health status.

## 1. Introduction

Across the world, slow-growing male layer chicks are a waste of the egg industry, with more than 7 billion being culled after hatching every year. Currently, two alternatives are proposed to solve this ethical problem: The use of several methods to determine the sex of the egg before hatching and the rearing of the males to produce meat. The former involves techniques (the penetration of the egg with a needle, the requirement to open the eggshell) that are risky and invasive [1]. Besides, in vivo sex determination is mostly available in a few developed countries (Germany, France, Netherlands, etc.). Thus, there is a growing interest in using the latter, as demonstrated in numerous studies [2,3,4,5,6]. Although male egg-type chicks, from a health and welfare perspective, are good for rearing in alternative production systems (rearing with outdoor access) as they have shown more active behavior, spent more time outdoors, and had better bone quality [7,8], it is challenging to fatten them [9]. For instance, in a recent investigation, Evaris et al. [10] found that slow-growing male chickens, aged 98 and 115 days old that were provided daily range access for 97 days, had a feed conversion rate of 5.88 and 7.71, respectively, which implies that some productive performance values (particularly feed conversion) were outside the expected range of valid standards during the finisher stage of these birds’ life. Targeting this specific phase is essential to assure the worth of raising them up to 18 weeks of age.

Dominant Blue D 107 chicks are a cross between the Blue Plymouth Rock rooster and Barred Plymouth hens. They are suitable for production systems with outdoor access. Hens are used to produce eggs, and males could be raised to produce meat; however, the latter is not well-studied. For example, with livability of up to 97%, at 18 weeks old, the males are expected to reach 2100 g for feed consumption of 6800 g [11]. Even though experiments with this specific breed are scarce, in a trial with male Dominant CZ chickens that lasted 16 weeks, Ibrahim et al. [12] reported a body weight of 1821 g and 1567 g, an overall feed intake of 6515.6 g and 6434.4 g, and a feed conversion of 3.67 and 4.28 for Dominant Red Barred and Dominant Sussex, respectively. The improvement in growth performance and carcass quality parameters, which depends on factors such as strain, sex, production system and management, environment, climate, slaughter age and weight, nutrients digestion, absorption and metabolism, feed wastage, water intake, dietary energy, protein, fiber, fat, minerals, and vitamin A, B, C, and D levels, is key to promoting the rearing of these birds.

On the other hand, in recent decades, the ban on the use of antibiotics as growth promoters in animal feeding to help fatten livestock has stimulated research exploring new and natural sources for enhancing growth [13]. Recently, phytogenic feed additives or phytobiotics, which include plants, plant extracts, herbs, spices, ether or essential oils, and resins, have shown promising results in improving the growth performance of farm animals [14,15]. They have antioxidant, anti-inflammatory, antimicrobial, antibacterial, anthelmintic, coccidiostatic, antiviral, and immunomodulating properties [16], through bioactive compounds such as vitamins, minerals, flavonoids (myrecytin, quercetin, kaempferol), phenolic acids (chlorogenic and caffeic acids), carotenoids, alkaloids, glucosinolates, isothiocyanates, and saponins [17,18] that can stimulate the secretion of digestive enzymes, improve feed’s flavor and palatability, activate feed intake, and enhance gut health [16]. According to Basit et al. [19], some leaf meals have most of the properties previously mentioned due to their secondary metabolites content. 

Recently, there has been a growing interest in using *Moringa oleifera* for poultry feeding. Available in most tropical and subtropical countries, including Mexico, dry *Moringa oleífera* leaves (MOL) are rich in nutrients such as protein (22.9–29.36%), ash (8.05–10.38%), carbohydrates (47.25–56.28%), crude fiber (6–9.6%), and fat (4.03–9.51%) that are good for poultry health and development [20]. MOL’s amino acid profile is also impressive as most essential and nonessential ones, namely valine (1.23–1.44%), leucine (1.69–1.94%), isoleucine (1.12–1.3%), histidine (0.53–0.62%), lysine (1.24–1.39%), methionine (0.33–0.42%), phenylalanine (1.41–1.5%), tryptophan (0.88–1.02%), threonine (1.33–1.42%), glycine (8.74–9.4%), alanine (0.98–1.09%), proline (1.04–1.19%), arginine (1.43–1.55%), cysteine (0.37–0.44%), serine (0.79–0.92%), aspartic (1.29–1.46%), and glutamic (2.33–2.45%), are present [21]. In addition to antioxidant and immunomodulating activities, MOL’s content of phytochemicals, bioactive compounds, and secondary metabolites such as phytates (2.57%), trypsin inhibitors (3.00%), vitamin E (0.011–0.077%), vitamin C (0.016–0.017%), vitamin A (0.016–0.019%), calcium (2.185–3.050%), phosphorus (0.204–0.252%), potassium (1.236–1.384%), saponins (1.60%), tannins (1.32–2.06%), oxalates (0.45%), quercetin (0.1%), cyanide (0.1%), and isothiocyanates (1.66%) [18,20,22,23] make it a potent natural feed additive or feed ingredient as a supplement suitable for poultry dietary formulations to improve growth performance through enhanced gut health [24]. However, the presence of certain secondary metabolites such as polyphenols (tannins) may negatively affect nutrient digestibility [25] and absorption in poultry, resulting in higher feed conversion. For instance, Cui et al. [26] observed a decline in some parameters of productive performance with increasing dietary levels of *Moringa oleifera* leaf meal (MOLM) in broiler chickens and recommended up to 1.56% inclusion in the diets. Regarding slow-growing birds, Sebola et al. [27] suggested up to 7% of Moringa in the diet of indigenous chickens intensively grown.

Although a range between 1.56 and 7% Moringa inclusion has been recommended for chicken diets, considering that chicken breeds could react differently to any given vegetable material, it is important to investigate the effects of specific plant species, inclusion rates, animal type, and life stages (i.e., layer, broiler, slow-growth, starter, grower, finisher) at different locations (temperate, tropic). Moreover, data on finisher slow-growing breeds’ performance when fed Moringa while having access to outdoors in the tropics are limited. Thus, this study was designed to evaluate the effects of diets supplemented with different levels of *Moringa oleifera* meal (MOM) on productive performance, carcass characteristics, and blood parameters of finisher egg-type slow-growing Dominant CZ Blue D 107 male chickens grown with outdoor access in a tropical environment.

## 2. Materials and Methods

### 2.1. Bird Management and Experimental Diets

This study was carried out in a half-block and half-wire mesh-sided house with a concrete floor and natural ventilation at the Faculty of Veterinary Medicine and Animal Science (FMVZ) of the University of Yucatan (UADY) in Mexico and followed all bioethics requirements of the Biological and Agricultural Sciences Campus (CCBA) of the UADY (CB-CCBA-S-2022). The climate of the region is warm and subhumid. According to INEGI [28], the average annual temperature is 26 °C, with relative humidity and average annual rainfall reaching 72% and 1100 mm, respectively. Three hundred and six 1-day-old layer-type slow-growing Dominant CZ Blue D 107 chickens arrived at FMVZ-UADY. All birds, reared in a spacious floor pen, received commercial feed (21% CP and 2.9 Mcal/kg ME) until 72 d-old and were vaccinated against Gumboro and Newcastle diseases at 7 and 21 d of age.

Since, in a previous trial, Evaris et al. [5] found that some productive traits of finisher slow-growing male chickens raised with outdoor access in the tropics were out of range (higher than usual), the present investigation targeted only the finisher stage of the birds. Thus, when the birds turned 72 d-old, one hundred and ninety-eight males were selected. They had an average live weight of 1093 ± 15.7 g and were randomly allocated to three dietary treatments that consisted of 0 (control or T1), 30 (T2), and 60 (T3) g/kg of *Moringa oleifera* meal (MOM) inclusion in a finisher diet, which was balanced to contain 2.9 Mcal/kg of metabolizable energy and 17% of crude protein according to the nutrient requirements of slow-growing birds raised under tropical conditions reported by Baas-Osorio et al. [29]. Chickens, housed in 18 floor pens (2.3 m × 1 m, each) equipped with feeders and drinkers and shaving wood as bedding, received experimental finisher diets and water *ad libitum* and natural daylight. Each treatment was subdivided into 6 individual pens (or replicates) of 11 animals. According to their replicates, birds had daily outdoor access (8 a.m. to 5 p.m.) to 18 pasture area courts (11.0 m × 1 m, each) via a doorway (50 cm × 70 cm, each) until the end of the experiment, which lasted for 7 weeks (49 days). The pasture area was covered with tropical natural vegetation, mainly buffelgrass (*Cenchrus ciliaris*).

Fresh Moringa leaves and edible green stems were collected at FMVZ-UADY and a nearby location, dried in an oven at 60 °C for 72 h, ground (3-mm mesh), and added in proportions of 30 and 60 g/kg to the diets. The MOM inclusion levels were selected according to the previously published recommendations ranging from 1.56% in broilers [26] to 7% in slow-growing breeds [27]. The chemical composition of MOM is shown in Table 1. The procedures of AOAC International [30] were used to analyze crude protein, ether extract, and ash of MOM. Crude fiber (ANKOM method no. 7), neutral detergent fiber (ANKOM method no. 6), acid detergent fiber (ANKOM method no. 5), and lignin (ANKOM method no. 8) were analyzed according to Van Soest et al. (1991) [31]. Total phenols (Folin–Ciocalteu method), total tannins (adding polyvinyl polypyrrolidone), and condensed tannins (vanillin method) were measured as reported by Ortíz-Domínguez et al. (2021) [32].

The ingredients and chemical compositions of the experimental diets are shown in Table 2. However, it is worth mentioning that the chemical analysis of the diets revealed that the values for crude protein and crude fiber were different from the calculated ones. Feed samples were analyzed using the methods of NMX [33] for dry matter, and AOAC International [30] for crude protein and fiber. Due to budgetary restrictions, the full WEENDE analysis was not performed, which constitutes a ‘‘limitation of the study’’ to declare the MOM effect on the birds with total confidence.

### 2.2. Growth Performance Evaluation

Data for the live weight (LW) were collected at the start (when birds were 72 d-old), halfway (when birds were 99 d-old), and at the end of the experiment (when birds turned 120 d-old). Live weight gain (LWG) was the difference between the current and previous LW. Daily live weight gain (DLWG) was obtained by dividing LWG between the number of days from 2 weighing periods. Feed intake (FI) was measured as the difference between the offered and rejected feed weight every 72 h. Daily feed intake (DFI) was the ratio of the FI and the number of feeding days elapsed. The feed conversion rate (FCR) was calculated as the ratio of FI and LWG × 100. The mortality rate (MR) was estimated as the percentage of birds that died per period. While FI, LW, and LWG are reported as grams per bird (g/b), DFI and DLWG are expressed as grams per bird per day (g/b/d). All productive performance data are reported for the first 4 weeks or the first period (72–99 d of age), the last 3 weeks or the last stage (100–120 d), and for the whole experiment or cumulative (72–120 d).

### 2.3. Carcass Characteristics Measurement

One day prior to slaughtering, birds’ weight was recorded, and feed was withheld for 12 h with access to water. At 121 d of age, 4 birds per pen (24 animals/treatment) were selected at random, and each one of them was humanely sacrificed by hand, scalded (60 °C for 3 min), and plucked with an automated machine. After that, carcasses were eviscerated, prechilled (4 °C for 1 h), and weighed for carcass weight and yield. The weight of abdominal fat, gizzard, liver, heart, and feet was also recorded. Each gizzard was emptied and cleaned before registering the weight. Subsequently, after being chilled (4 °C for 18 h), each eviscerated carcass was cut into parts (2 wings, 2 legs, 1 breast), and the weight of each part was registered without removing the skin and bone. Expressed as a percentage, breast yield, leg yield, wing yield, feet yield, gizzard yield, liver yield, and heart yield were estimated on both the final live weight (FLW) and eviscerated carcass weight (ECW) basis.

### 2.4. Blood Sampling Assessment

From the 24 birds/treatment selected for carcass trait evaluation, 2 birds/pen (12 animals/treatment) were randomly selected for blood sampling before slaughtering. Thus, samples of 2 mL of blood were collected in sterilized tubes with and without Ethylenediaminetetraacetic acid (EDTA) via wing veins from each bird. Samples with and without anticoagulants were stored at 4 °C and −20 °C, respectively, until biochemical analyses at the FMVZ laboratory. Blood trait determination followed the procedure described by Hernandez et al. [34]. From each sample, total protein, albumin, and alkaline phosphatase were determined by the colorimetric method using the CHEMILYZER-HLAB equipment with the Accutrack brand kit. Red blood cells or erythrocytes and white blood cells or leukocytes (monocytes, eosinophils, heterophils, and lymphocytes) were determined by manual counting (Neubauer camera, Olympus microscope, Natt and Herrick diluent solution). The cyanmethemoglobin method (Drabkin’s reagent, Hycel lab equipment) was used to determine hemoglobin. Hematocrit was analyzed by microhematocrit (Microcentrifuge M-08F). Globulin was calculated by subtracting albumin from the total protein. The mean corpuscular volume (MCV) was obtained from the hematocrit and the erythrocyte count using the formula MCV = Hct × 10/Red blood cells, while the mean corpuscular hemoglobin concentration was calculated from the hematocrit and hemoglobin using the formula MCHC = Hb × 100/Hct, where Hb is Hemoglobin and Hct is Hematocrit.

### 2.5. Statistical Analyses

The data were analyzed by a one-way ANOVA using the statistical software Minitab 19 [35]. Since the LW of the selected birds for sacrifice was different between treatments, it was used as a covariate to analyze carcass traits. Means of treatment were compared using the Tukey test, and *p*-values ≤ 0.05 were considered significantly different. No statistical analysis was performed for mortality rate and abdominal fat, as most values were nil.

## 3. Results

### 3.1. Diet Effects on Growth Traits

The effects of diets supplemented with 0, 30, and 60 g/kg MOM on the productive performance of slow-growing Dominant CZ Blue D 107 male chickens raised with outdoor access in a tropical environment are shown in Table 3 and Table 4.

#### 3.1.1. Average Live Weight (LW)

At the start of the trial, LW between treatments was not statistically different as birds from T1, T2, and T3 had an average weight of 1099.92 g, 1080.83 g, and 1100.14 g, respectively. However, after the first 4 weeks of the experiment, birds that were fed the control diet (T1) had higher LW (*p* ≤ 0.05) than the other two groups (T2 and T3). At the end of the study, treatments did not have a significant effect on the animals’ LW even though birds from T1 were numerically heavier, followed by T3 and T2, respectively.

#### 3.1.2. Average Live Weight Gain (LWG) and Daily Live Weight Gain (DLWG)

LWG was significantly different (*p* ≤ 0.05) between treatments throughout the experiment. During the first 4 weeks of the trial, birds fed diets with 30 and 60 g/kg MOM gained 12.86% and 16.61% less weight, respectively, than those raised on the diet without MOM. However, in the last 3 weeks of the experiment, a significant increase in LWG was observed in birds from T2 (11.13%) and T3 (14.84%) in comparison to animals from T1. In contrast to LWG per period or stage, cumulative LWG was not influenced (*p* ≥ 0.05) by the treatments. At the end of the trial, birds raised without Moringa gained a total of 1118.18 g, while the ones grown with 30 and 60 g/kg MOM added 1083.60 g and 1074.40 g, respectively, to their initial weight. DLWG followed the same pattern as LWG (Table 3).

#### 3.1.3. Average Feed Intake (FI), Daily Feed Intake (DFI), Feed Conversion Ratio (FCR), and Mortality Rate (MR)

FI did not differ between groups (*p* ≥ 0.05) during the first 4 weeks and for the whole experiment (cumulative FI). However, in the last 3 weeks of the trial, birds that were fed dietary MOM increased their FI (*p* ≤ 0.05) compared to those fed the diet without MOM. There was no difference (*p* ≥ 0.05) between treatments for DFI throughout the experiment. Compared to the animals that were fed the MOM-free diet, cumulative FCR, as well as FCR for the first 4 weeks, were higher (*p* ≤ 0.05) in birds fed the dietary MOM inclusion. Treatments did not have a significant effect on FCR during the last stage of the trial, which corresponds to the period where the birds had their highest FCR. MR amounted to 7.58% as 5 birds from T3 died in the last three weeks of the study (Table 4).

### 3.2. Diet Effects on Slaughter Performance

The results of carcass characteristics are presented in Table 5. There were no statistical differences (*p* ≥ 0.05) between treatments for eviscerated carcass weight (ECW) and eviscerated carcass yield (ECY), with T3 having higher numerical values followed by T1 and T2, respectively. Birds, across all 3 treatments, had no measurable abdominal fat. Breast weight and breast yield, on both final live weight (FLW) and ECW bases, were significantly higher (*p* ≤ 0.05) in birds fed the dietary inclusion of MOM compared to those that were fed the control diet. Even though leg weight and leg yield, along with wing weight and wing yield, were not influenced by the treatments (*p* ≥ 0.05), birds that received T3 had greater numerical values than those from T1 and T2. However, birds’ feet from T1 were significantly heavier and yielded more on an FLW basis than chickens’ feet from T2 and T3, separately. The gizzard weight and yield, on an ECW basis, were significantly greater (*p* ≤ 0.05) in birds fed Moringa compared to the ones fed the control diet. Liver weight and yield, as well as heart weight and yield, were not affected by the treatments (*p* ≥ 0.05).

### 3.3. Diet Effects on Blood Characteristics

Table 6 presents the blood test results of Dominant CZ Blue D 107 male chickens fed a dietary inclusion of 0, 30, and 60 g/kg MOM, while having access to tropical outdoors. Supplementation of MOM in the diet did not significantly alter (*p* ≥ 0.05) the parameters used to evaluate the impact of Moringa on the birds’ health status.

## 4. Discussion

### 4.1. Diet Effects on Growth Performance

Most available studies have evaluated the effects of Moringa on broilers; thus, the results of the current study are also compared with those of both fast- and slow-growing chickens, as the outcomes reported in the literature vary. For instance, Kumar et al. [36] demonstrated that incremental dietary MOLM inclusion resulted in lower live weight (LW) in Vanaraja chickens, which is a slow-growing breed. Conversely, Alshukri et al. [37] reported that the final LW increased with increasing MOLM amounts in broiler chicken diets. In this study, during the first 4 weeks of the trial, LW was significantly lower in birds that had Moringa in their diets than in those that had the control one. However, the statistical differences disappeared in the last 3 weeks of the trial. Differences in results between periods could be attributed to adaptation to MOM in relation to dietary fiber levels and content of phytochemicals, bioactive compounds, and secondary metabolites. It should be noted that, in this study, birds from all 3 treatments reached the threshold of 2100 g before 18 weeks of age. This feat implies that MOM did not negatively influence the bird’s LW.

As for live weight gain (LWG) and daily live weight gain (DLWG), Gadzirayi and Mupangwa [38] found that the former trait was significantly lower with diets supplemented with increasing levels of MOLM in indigenous chickens; however, Gakuya et al. [39] observed no effects of the increase in dietary MOLM inclusion on the same trait in layer chickens. Ayssiwede et al. [40] evaluated the effects of 0% and 8% MOLM on indigenous Senegal chickens and reported a DLWG of 6.49 g and 8.77 g, respectively, which is 3-fold less than that reported in the current experiment. LWG and DLWG per stage, in this trial, followed an unexpected trend as birds fed on the control diet gain significantly more weight than those that had MOM in their diets during the first 4 weeks, but considerably gain less weight in the last 3 weeks of the trial. As unpredicted as this could be, Etienne et al. [41] observed the same trend in traditional chickens in Northern Côte d’Ivoire that were fed with 0, 5, and 10% Moringa in their diets. These authors reported that chickens’ DLWG improved with 5% dietary Moringa at the end of their trial. This improvement in LW and DLWG might be related to some other factors that were not measured in the corresponding experiments. For instance, MOM has isothiocyanates (mostly moringin or glucomoringin), which have powerful anti-inflammatory, antibacterial, and antioxidant properties [42,43] that might improve LW over time, consequently influencing LWG and DLWG. Furthermore, Divya et al. [24] found that broilers’ gut health was improved with Moringa; thus, Moringa’s bioactive compounds with antimicrobial activity and immune-stimulating actions might have led to better use and absorption of nutrients at the end of the current study. Furthermore, the adaptation of slow-growing chickens to dietary fiber after some period of time might have contributed to these results.

Several studies have revealed that dietary MOLM inclusion does not significantly affect the feed intake (FI) [37,39,41,44]; however, others detected significantly lower FI with increasing MOLM levels in chickens’ diets [38], and still others observed higher FI with MOLM inclusion [36,45]. In the current study, FI increased with MOM levels only during the last 3 weeks of the trial as the overall or cumulative FI along with FI for the first 4 weeks were not significantly different. The increased FI in the last 3 weeks of the trial might be attributed to the function of MOM as a feed stimulus by improving the feed’s flavor, smell, and palatability, which might have influenced eating patterns and activated feed intake and secretion of digestive fluids in the animals [11]. As for daily feed intake (DFI), Melesse et al. [46] reported that it was augmented with increasing dietary levels of *Moringa stenopetala* in slow-growing South African Koekoek chickens. In contrast, Ayssiwede et al. [40] noticed that DFI decreased with higher rates of MOLM inclusion in the diets of indigenous Senegal chickens. However, in the current study, treatments did not significantly affect DFI. Even though this result aligns with the findings of Etienne et al. [41], who registered no meaningful MOLM effects on DFI of traditional chickens in Côte d’Ivoire, it differs in that these authors reported values that are well below the data of our study. Differences might be due to the different biological materials, as well as the composition of the diets used in each experiment. The results of the current experiment for FI and DFI imply that up to 60 g/kg MOM, as a feed ingredient, could be included in these birds’ diets and likely contribute to reducing the feeding cost, which amounts to up to 70% of the total cost of production [47,48]. As shown in Table 2, less corn meal and soybean were used with the inclusion of MOM.

When it comes to the feed conversion rate (FCR), it indicates how the bird converts feed intake into live weight. This trait, when substantially high, has a negative impact on production cost. Variable results have been previously registered for this trait. For instance, Sebola et al. [27] reported lower FCR in slow-growing Ovambo chickens that were fed without MOLM in comparison to those that received incremental levels of MOLM. However, Etienne et al. [41] registered a lower FCR in birds fed Moringa than in those fed a diet without Moringa. Ayssiwede et al. [40] found that dietary supplementation with MOLM had no significant effect on the FCR of indigenous Senegal chickens. In the current study, FCR was significantly higher in the animals that received dietary MOM inclusion than in those that did not, both during the first 4 weeks of the experiment and for the cumulative period. This result was expected as slow-growing birds are the biological material used in this experiment, and it has been demonstrated that this genotype has a high FCR [2,5]. Furthermore, MOM’s secondary metabolite content such as flavonoids and phenolic compounds may have played a role in this result [25] since Cui et al. [26] reported higher FCR with increasing MOLM levels in broilers due to low digestibility. In fact, total phenols, total tannins, and condensed tannin contents of the MOM used in the current study were 2.78%, 0.92%, and 3.27%, respectively. When consumed, these antinutritional factors may have had a negative impact on the availability of nutrients for the birds. However, it is important to note that FCR values for the present study at 99 and 120 days old are well below those found in 98- and 115-day-old Rhode Island Red slow-growing chickens raised with outdoor access for 97 days without MOM in their diets [5]. This suggests that the association of MOM and a shorter outdoor access period (only 49 days in this trial) might have contributed to lowering the FCR of these birds. Since FCR is influenced by genetic, nutritional, production system, and management factors, reducing feed wastage may help improve this trait [2]. When it comes to MR, it represented 7.58% of birds fed T3 in comparison to 0% for the other groups, without statistical analyses to evaluate the significance. It is essential to note that these birds died during the last 3 weeks of the trial due to outdoor predators, which is one of the disadvantages of raising birds with access to the outdoors. Special infrastructure is needed to protect animals from predatory exposure, such as dogs, coyotes, foxes, raccoons, and snakes [49,50].

### 4.2. Diet Effects on Slaughter Parameters

Similar to growth performance, the literature differs regarding the effects of dietary supplementation of Moringa on chicken carcass traits [51,52,53]. In our study, carcass weight and carcass yield were not significantly influenced by the treatments, a result that agrees with some authors [40,54,55,56] but contrasts with the findings of others [36,46]. Discrepancies between studies could be attributed to the different biological materials and diets used in each experiment, as well as the climate and environment where the trials took place, which in turn may affect both the animal performance and the Moringa composition.

Alternatively, the absence of abdominal fat might be due to the fact that the birds used in this experiment are slow-growing layer-type male chickens that had access to the outdoors. It has been demonstrated that these layer male birds, when raised with outdoor access, deposit less abdominal fat [5].

On the other hand, the inclusion of MOM ensued in an increase in breast weight and yield. This is a tremendous outcome as the breast muscle is the most valuable part of the bird [3]. These results could be attributed to the amount and quality of amino acids from MOM. Breast muscle has more protein than fat; consequently, a surplus of amino acids from MOM may have increased protein synthesis in this muscle type and promoted its development. In fact, Mahfuz and Piao [17] stated that higher breast muscle weight could be assigned to the increased protein deposition in birds fed diets supplemented with Moringa. Considered as one of the most important benefits of MOM to chicken feeding in our study, this improvement in breast weight could also be the result of the high content of phytobiotics from Moringa, since Krauze [57] indicated that phytobiotics may increase the weight of the pectoral muscle in chickens. Finally, this higher breast weight could be associated with the fiber diameter of this muscle, as Rehman et al. [58] stated that Moringa increased breast muscle fiber diameter.

Dietary inclusion of Moringa had no significant effect on leg (thigh and drumstick) weight and leg yield, as well as wing weight and wing yield in the current trial. However, Melesse et al. [46] found significantly higher thigh, drumsticks, breast, and wing weight and yield in slow-growing South African Koekoek chickens that were fed a dietary inclusion of *Moringa stenopetala.* Nevertheless, these authors also reported a significantly higher slaughter weight without a covariate analysis, which might have influenced the statistical significance.

On the other hand, chicken feet are not a waste material from the meat industry as they are consumed in many countries, transformed into meal to be incorporated into animal feeding, processed into ready-to-use dried chicken soup, and are a good source of high-quality gelatin and collagen [59,60]; thus, chickens’ feet weight matters. In this study, chicken feet from T1 had higher weight and yield than those from T2; however, these parameters did not differ statistically between T1 and T3. This implies that high chicken feet weight and yield can also be achieved with 60 g/kg MOM.

Ultimately, gastrointestinal tract development and functionality in chickens are of utmost importance since they may affect feed efficiency. For instance, the gizzard is a vital organ for feed grinding and nutrients and digestive fluid mixing [61]. While several authors have observed significant effects of MOLM on broilers’ gizzard, liver, or heart weight and yield [52,62,63,64], others found no MOM effects on the same internal organs [65]. The non-significant MOM effect on the heart and liver in the current study is supported by reports of Sebola and Mokoboki [66] and Ayssiwede et al. [40], who observed no effect of MOLM on such organs of indigenous chickens. In the present trial, the gizzard was heavier and yielded more in birds fed MOM than in those fed the control diet. This result could be explained by the presence of dietary fiber from Moringa, which may have lasted longer in the gizzard, prompting the development of its muscles that in turn ensured proper feed grinding, intensified intestinal refluxes, and enabled the blending of the digesta with digestive enzymes; thus, resulting in enhanced digestion and better nutrient efficiency [67].

### 4.3. Diet Effects on Blood Traits

Biochemical and hematological profiles are used to evaluate the functionality of many critical systems and organs, such as the liver, heart, spleen, kidney, gizzard, and pancreas, among others. Total protein, which comprises albumin and globulin, provides amino acids to the tissues, and supports the immune system of the body. Alkaline phosphatase (ALP), which is a liver enzyme, regulates bile flow in the liver. Leukocytes (monocyte, eosinophil, lymphocyte) are white blood cells that produce antibodies to help the system fight off infection due to bacteria, viruses, and fungi, while erythrocytes (red blood cells) transport oxygen from the lungs to the tissues and carry carbon dioxide back to the lungs. Even though values for total protein, albumin, lymphocytes, and erythrocytes were numerically improved in birds raised with 6% MOM, and animals from T2 had greater numerical values of globulin, ALP, MCV, and eosinophils, whereas birds fed without Moringa registered a higher amount of hemoglobin, hematocrit, MCHC, and leukocytes in their blood as their number of heterophils and monocytes were equivalent to those from chickens grown with the T2 diet, the data of the current experiment indicated that there were no dietary effects on blood characteristics. These results align with those found by Ashour et al. [68] and Zanu et al. [55], who reported no effects of Moringa leaf and/or seed meal on some blood traits (albumin, total protein, globulin, MCV, MCHC) of Japanese quails and broiler chickens, respectively. Contrary to our results, Sebola and Mokoboki [66] and Abbas et al. [69] showed that MOLM substantially affected the blood of indigenous male chickens and broilers, correspondingly. Variability between current and previous results might be due to environmental, geographical, nutritional, and genetic changes. In this trial, the absence of significant effects of MOM on blood parameters may suggest that the birds had healthy internal organs.

## 5. Conclusions

In spite of its limitations (lack of the full WEENDE analysis for MOM and experimental diets), the results of this study provide insights into the usefulness of MOM in the diet of finisher slow-growing Dominant CZ Blue D 107 male chickens raised with outdoor access. The inclusion of 30 g/kg MOM resulted in higher breast weight and yield and greater gizzard weight and yield. Therefore, in a tropical environment, as a feed ingredient, up to 60 g/kg MOM could be included in these birds’ diets without adverse effects on their biological performance (all traits but feed conversion), carcass quality, and health status.

## Figures and Tables

**Table 1 animals-12-01843-t001:** Chemical analysis of MOM (dry matter basis).

Chemical Analysis (%)	MOM
Crude Protein (CP)	19.11
Crude Fiber (CF)	15.39
Neutral Detergent Fiber (NDF)	48.78
Acid Detergent Fiber (ADF)	25.43
Ether Extract (EE)	2.97
Ash	8.74
Lignin	5.70
Condensed Tannins	3.27
Total Tannins	0.92
Total Phenols	2.78

**Table 2 animals-12-01843-t002:** Composition, calculated values, and chemical analysis of the experimental diets (in percent).

Ingredients (%)	T1	T2	T3
MOM	0	3	6
Corn meal	69.96	66.48	64.10
Soybean meal (44% CP)	25.11	24.47	23.55
Choline chloride	0.05	0.05	0.05
Calcium carbonate	2.31	2.50	2.53
Calcium orthophosphate	1.70	2.64	2.90
Salt	0.25	0.25	0.25
Lysine	0.28	0.27	0.27
DL-Methionine	0.16	0.16	0.17
Vitamins premix ^1^	0.10	0.10	0.10
Minerals premix ^2^	0.08	0.08	0.08
Total	100	100	100
Calculated values			
Metabolizable Energy (Mcal/kg)	2.94	2.91	2.90
Crude Protein (%)	17.00	17.01	17.00
Crude Fiber (%)	3.30	3.99	4.68
Lysine (%)	1.13	1.13	1.13
Methionine (%)	0.44	0.49	0.55
Methionine + Cystine (%)	0.73	0.73	0.73
Calcium (%)	1.34	1.68	1.81
Available phosphorus (%)	0.68	0.85	0.89
Chemical analysis (%) dry matter base			
Dry matter	87.35	87.56	88.05
Crude Protein	21.01	20.17	21.75
Crude Fiber	1.99	2.57	2.66

^1^ Minerals Premix (mg/kg of diet): Manganese,12; iodine, 1.25; iron, 50; copper, 16; zinc, 100; selenium, 0.3; ^2^ Vitamins Premix (unit/kg of diet): vitamin A, 12,000 UI; vitamin D3, 3500 UI; vitamin E, 50 UI; vitamin K3, 4 g; vitamin B1, 3 g; vitamin B2, 10 g; vitamin B6, 6 g; vitamin B12, 18 mg; biotin, 250 mg; folic acid, 2 g; niacin, 60 g; pantothenic acid, 18 g; antioxidant, 125 g.

**Table 3 animals-12-01843-t003:** Live weight (LW), live weight gain (LWG), and daily live weight gain (DLWG) of slow-growing Dominant CZ Blue D 107 male chickens raised with *Moringa oleifera* meal (MOM) dietary inclusion in a tropical environment.

Traits	T1 (0 g/kg MOM) (Mean ± SEM *)	T2 (30 g/kg MOM) (Mean ± SEM)	T3 (60 g/kg MOM) (Mean ± SEM)	*p*-Value
LW (g/b)				
72 d	1099.9 ± 14.0	1080.8 ± 19.1	1100.1 ± 14.1	0.616
99 d	1766.7 ^a^ ± 18.1	1661.9 ^b^ ± 25.3	1658.2 ^b^ ± 18.6	0.001
120 d	2218.1 ± 22.4	2164.4 ± 33.3	2176.6 ± 24.8	0.341
LWG (g/b)				
72–99 d (first 4 weeks of trial)	666.8 ^a^ ± 8.4	581.0 ^b^ ± 12.6	556.0 ^b^ ± 8.6	0.001
100–120 d (last 3 weeks of trial)	451.4 ^b^ ± 9.9	502.6 ^b^ ± 12.7	518.4 ^a^ ± 12.5	0.001
Cumulative (72–120 d)	1118.2 ± 13.9	1083.6 ± 19.5	1074.4 ± 16.4	0.153
DLWG (g/b/d)				
72–99 d	23.8 ^a^ ± 0.3	20.8 ^b^ ± 0.5	19.9 ^b^ ± 0.3	0.001
100–120 d	22.6 ^b^ ± 0.5	25.1 ^a^ ± 0.6	25.9 ^a^ ± 0.6	0.001
Cumulative (72–120 d)	23.3 ± 0.3	22.6 ± 0.4	22.4 ± 0.3	0.153

^ab^ Means with different letters are significantly different (*p* ≤ 0.05). * SEM: Standard error of mean.

**Table 4 animals-12-01843-t004:** Feed intake (FI), daily feed intake (DFI), feed conversion rate (FCR), and mortality rate (MR) of slow-growing Dominant CZ Blue D 107 male chickens raised with *Moringa oleifera* meal (MOM) in a tropical climate.

Traits	T1 (0 g/kg MOM) (Mean ± SEM *)	T2 (30 g/kg MOM) (Mean ± SEM)	T3 (60 g/kg MOM) (Mean ± SEM)	*p*-Value
FI (g/b)				
72–99 d	2677.2 ± 29.8	2604.3 ± 66.6	2512.0 ± 114.7	0.354
100–120 d	1958.4 ^b^ ± 34.7	2285.4 ^a^ ± 16.4	2384.2 ^a^ ± 37.4	0.001
Cumulative (72–120 d)	4635.6 ± 45.2	4889.7 ± 74.4	4896.2 ± 148.2	0.140
DFI (g/b/d)				
72–99 d	97.0 ± 1.2	94.0 ± 2.9	90.6 ± 4.4	0.373
100–120 d	97.6 ± 1.7	114.4 ± 0.9	112.1 ± 8.7	0.072
Cumulative (72–120 d)	97.3 ± 1.1	104.2 ± 1.6	101.3 ± 6.3	0.456
FCR				
72–99 d	4.0 ^b^ ± 0.0	4.5 ^a^ ± 0.2	4.5 ^a^ ± 0.2	0.016
100–120 d	4.3 ± 0.1	4.6 ± 0.2	4.7 ± 0.2	0.247
Cumulative (72–120 d)	4.2 ^b^ ± 0.0	4.5 ^a^ ± 0.1	4.6 ^a^ ± 0.1	0.001
MR (%) ^1^				
72–99 d	0.0	0.0	0.0	
100–120 d	0.0	0.0	7.6	
Cumulative (72–120 d)	0.0	0.0	7.6	

^ab^ Means with different letters are significantly different (*p* ≤ 0.05). * SEM: Standard error of mean. ^1^ MR in 60 g/kg MOM group was due to predator attack.

**Table 5 animals-12-01843-t005:** Effects of *Moringa oleifera* meal (MOM) on carcass characteristics of slow-growing Dominant CZ Blue D 107 male chickens raised with outdoor access in a tropical climate.

Traits	T1 (0 g/kg MOM) (Mean ± SEM *)	T2 (3 g/kg MOM) (Mean ± SEM)	T3 (6 g/kg MOM) (Mean ± SEM)	*p*-Value
FLW * (g)	2349 (covariate)	2349 (covariate)	2349 (covariate)	
ECW ** (g)	1540.6 ± 10.5	1515.2 ± 10.5	1542.5 ± 10.8	0.134
ECY *** (%)	65.6 ± 0.5	64.5 ± 0.5	65.7 ± 0.5	0.116
Abdominal Fat	0.0	0.0	0.0	
Breast Weight (g)	385.4 ^b^ ± 5.3	403.5 ^a^ ± 5.3	401.6 ^a^ ± 5.5	0.035
Breast Yield (% FLW)	16.4 ^b^ ± 0.2	17.1 ^a^ ± 0.2	17.1 ^a^ ± 0.2	0.045
Breast Yield (% ECW)	25.0 ^b^ ± 0.3	26.6 ^a^ ± 0.3	26.0 ^a^ ± 0.3	0.002
Leg Weight (g)	563.9 ± 4.3	559.5 ± 4.3	567.1 ± 4.4	0.484
Leg Yield (% FLW)	24.0 ± 0.2	23.8 ± 0.2	24.2 ± 0.2	0.487
Leg Yield (% ECW)	36.6 ± 0.2	37.0 ± 0.2	36.8 ± 0.2	0.449
Wing Weight (g)	208.1 ± 2.0	206.3 ± 2.0	213.2 ± 2.1	0.065
Wing Yield (% FLW)	8.9 ± 0.1	8.8 ± 0.1	9.1 ± 0.1	0.057
Wing Yield (% ECW)	13.5 ± 0.1	13.6 ± 0.1	13.8 ± 0.1	0.290
Feet Weight (g)	103.5^a^ ± 1.2	96.9 ^b^ ± 1.2	100.3 ^ab^ ± 1.2	0.001
Feet Yield (% FLW)	4.4 ^a^ ± 0.1	4.1 ^b^ ± 0.1	4.3 ^ab^ ± 0.1	0.001
Feet Yield (% ECW)	6.7 ^a^ ± 0.1	4.4 ^b^ ± 0.1	6.5 ^ab^ ± 0.1	0.032
Gizzard Weight (g)	51.4 ^b^ ± 2.3	59.4 ^a^ ± 2.3	57.0 ^a^ ± 2.4	0.041
Gizzard Yield (% FLW)	2.2 ± 0.1	2.5 ± 0.1	2.4 ± 0.1	0.057
Gizzard Yield (% ECW)	3.3 ^b^ ± 0.1	3.9 ^a^ ± 0.1	3.7 ^a^ ± 0.1	0.023
Liver Weight (g)	31.1 ± 0.8	30.0 ± 0.8	31.8 ± 0.8	0.325
Liver Yield (% FLW)	1.3 ± 0.0	128.0 ± 0.0	1.4 ± 0.0	0.358
Liver Yield (% ECW)	2.0 ± 0.1	2.0 ± 0.1	2.1 ± 0.1	0.064
Heart Weight (g)	9.8 ± 0.3	8.9 ± 0.3	9.2 ± 0.3	0.084
Heart Yield (% FLW)	0.4 ± 0.0	0.4 ± 0.0	0.4 ± 0.0	0.082
Heart Yield (% ECW)	0.6 ± 0.0	0.6 ± 0.0	0.6 ± 0.0	0.154

^ab^ Means that do not share similar letters are significantly different (*p* ≤ 0.05). * FLW: Final live weight; ** ECW: Eviscerated carcass weight; *** ECY: Eviscerated carcass yield.

**Table 6 animals-12-01843-t006:** Health status of slow-growing Dominant CZ Blue D 107 male chickens raised with *Moringa oleifera* meal (MOM) in a tropical climate.

Traits	T1 (0 g/kg MOM) (Mean ± SEM *)	T2 (3 g/kg MOM) (Mean ± SEM)	T3 (6 g/kg MOM) (Mean ± SEM)	*p*-Value
Total protein (g/dL)	3.3 ± 0.1	3.4 ± 0.1	3.4 ± 0.1	0.539
Albumin (g/dL)	1.7 ± 0.0	1.8 ± 0.1	1.9 ± 0.1	0.343
Lymphocytes (%)	48.5 ± 3.4	47.0 ± 4.7	51.7 ± 2.3	0.650
Erythrocytes (million/mm^3^)	1.9 ± 0.2	1.8 ± 0.1	2.0 ± 0.2	0.740
Globulin (g/dL)	1.5 ± 0.1	1.6 ± 0.1	1.5 ± 0.1	0.900
ALP * (UI/L)	992.9 ± 47.5	1129.3 ± 85.0	1059.6 ± 46.3	0.314
MCV ** (fl)	178.3 ± 15.2	184.9 ± 13.1	164.0 ± 9.4	0.505
Eosinophils (%)	3.1 ± 0.7	4.6 ± 0.8	3.5 ± 0.8	0.375
Hemoglobin (g/dL)	10.7 ± 0.3	10.5 ± 0.3	10.2 ± 0.3	0.396
Hematocrit (%)	31.8 ± 0.9	31.7 ± 0.7	30.6 ± 0.9	0.546
MCHC *** (g/dL)	33.7 ± 0.6	33.3 ± 0.3	33.2 ± 0.3	0.696
Leukocytes (/mm^3^)	26,916.7 ± 1083.4	25,500.0 ± 1515.0	25,000.0 ± 1614.3	0.618
Heterophils (%)	44.3 ± 3.3	44.3 ± 4.0	41.2 ± 2.3	0.743
Monocytes (%)	4.2 ± 0.5	4.2 ± 0.4	3.7 ± 0.4	0.678

*p*-values *≤* 0.05 are significantly different. * ALP: Alkaline phosphatase; ** MCV: Mean corpuscular volume *** MCHC: Mean corpuscular hemoglobin concentration.

## Data Availability

Data presented in this study are available on fair request from the authors.

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
