# Peer review of "Male Layer Chicken’s Response to Dietary Moringa oleifera Meal in a Tropical Climate"

_animals, 2022, doi:10.3390/ani12141843_

Round 1
Reviewer 1 Report
- The authors have studied the effect of Moringa oleífera on performance, processing weights, and blood traits of male layer chickens. Unfortunately, the study and manuscript present many critical flaws:
- Comment 1: The authors have studied Moringa oleífera to improve male layer chickens' response to optimize production costs. They recognize that most male chicks are discarded in the egg production industry for economic reasons. Indeed, many countries consider requiring the fertile eggs to be sexed to avoid discarding hatched male layer chicks. Authors have not supported how growing up male layer birds can be more convenient than sexing fertile eggs. Indeed, the approach they propose seems to be unreasonable. Therefore, the study lacks justification.
- Comment 2: It is expected for the readers not to be that interested in the study, based on the previous comment.
- Comment 3: The crude protein content of the diets (a key nutritional factor) is confounded with the treatments. A difference of about 8% in crude protein exists between treatments. Consequently, no results can be attributed to the experimental treatments because the experimental design provides no mechanism to control this confounding factor. Therefore, conclusions are not backed up.
- Comment 4: Another confounding factor likely existed: the actual metabolizable energy (ME) value of the diets. It is not reasonable for the three treatment diets to have the same ME if the higher the inclusion rate of moringa, the lower the inclusion of corn and soybean meal. No other ingredient in the diet has provided the ME to compensate for the reduction from corn and soy.
Author Response
-
The authors have studied the effect of Moringa oleífera on performance, processing weights, and blood traits of male layer chickens. Unfortunately, the study and manuscript present many critical flaws:
Comment 1: The authors have studied Moringa oleífera to improve male layer chickens' response to optimize production costs. They recognize that most male chicks are discarded in the egg production industry for economic reasons. Indeed, many countries consider requiring the fertile eggs to be sexed to avoid discarding hatched male layer chicks. Authors have not supported how growing up male layer birds can be more convenient than sexing fertile eggs. Indeed, the approach they propose seems to be unreasonable. Therefore, the study lacks justification.
More justification can be found in Lines 49-55.
Comment 2: It is expected for the readers not to be that interested in the study, based on the previous comment.
More arguments and references we added in Lines 49-55 and 498-507.
Comment 3: The crude protein content of the diets (a key nutritional factor) is confounded with the treatments. A difference of about 8% in crude protein exists between treatments. Consequently, no results can be attributed to the experimental treatments because the experimental design provides no mechanism to control this confounding factor. Therefore, conclusions are not backed up.
In Table 1, when we compare each treatment with the control, the difference in CP is less than 1%, which did not affect at all the bird’s growth performance. The cumulative productive performance results (like LW, LWG and DLWG) found in the trial support the fact that the diets had quite similar available nutrients. Therefore, based on the results, conclusions are backed up.
On the other hand, other studies have reported results where the CP and/or the ME of the treatments is different than the control by +/- 1%. For example, Sayed-Ahmed and Shaarawy (2019). Effect of feeding Moringa oleifera forage on productive performance of growing goat kids. Egyptian Journal of Sheep & Goat Sciences, Vol. 14, No. 1, P: 25 – 37.
Also, Annongu et al. (2014). Geo-Assessment of Chemical Composition and Nutritional Evaluation of Moringa oleifera Seeds in Nutrition of Broilers. Journal of Agricultural Science; Vol. 6, No. 4; Page 119-124, Published by Canadian Center of Science and Education.
Finally, Manyeula et al. (2019). Nutrient digestibility, haemo-biochemical parameters and growth performance of an indigenous chicken strain fed canola meal–containing diets. Tropical Animal Health and Production, 51, pages 2343–2350. https://doi.org/10.1007/s11250-019-01949-4.
Comment 4: Another confounding factor likely existed: the actual metabolizable energy (ME) value of the diets. It is not reasonable for the three treatment diets to have the same ME if the higher the inclusion rate of moringa, the lower the inclusion of corn and soybean meal. No other ingredient in the diet has provided the ME to compensate for the reduction from corn and soy.
Even though MO had lower ME than corn and soy, it provided both protein and energy to the treatments as it replaced only 3% and 6% of the diet. In fact, the In-vivo results support the fact that its nutrient contribution is valuable since all cumulative performance results were unaltered (excluding FCR). Any discrepancies in the diets should be reflected in the bird’s performance, which is not the case in this study. Similarity in nutrients availability in the diets, be it CP or ME, is demonstrated in the results found. Therefore, conclusions are valid.

Author Response
- In your introduction, the research background could be improved. It should add some gut microbiota research results related to Moringa oleifera Please reorganize
In line 102, the gut health of chicken is taken into account now, and a reference was added.
3) In your discussion, please elaborate a sentence highlighting benefits of Moringa oleifera Meal to chickens.
In line 399, a sentence about the benefits of MOM to chickens was added.
Minor comments (examples rather than an exhaustive list of corrections):
Materials and Methods
Line 165: please change “productive” to “growth”
In Line 166, “Productive” has been replaced by “Growth”
Table 1: why methionine level among treatments were not the same? Please check.
In Table 1, the same amount of methionine was added to the diet; however, when balancing, it slightly increased with increasing Moringa levels because either methionine or methionine + cystine would stay unbalanced. So, we decided to keep methionine + cystine balanced.
Discussion:
4.1 Diets effects on growth performance
It would be better to describe that Moringa oleifera Meal changed the structure of gut microbiota and is associated with improved growth performance from some references.
Unfortunately, we did not directly evaluate gut health in this study. However, in Lines 319-325, we associated improved productive performance to enhanced gut health based on references found in the literature.
It’s interesting that diet supplementary Moringa oleifera Meal increased breast weight and yield. But the explanation of this phenomenon is not well, Usually, amino acids and protein from soybean meal are one of the most digestive, compared with other protein materials, please improve it.
In Lines 403-405, more explanation has been added.
Conclusion:
This part summarized not well, It’s too long, please shorten.
The conclusion has been made shorter
Tables:
Please use letter in accordance with all significant data. For example, bigger number add letter a, and smaller number add letter b.
In the Tables of results, bigger number has the letter a while smaller gets the letter b.

Reviewer 3 Report
In the simple summary and rest of the paper please change the word “killing” by a more professional word such as “disposed, or euthanized…”
Do you have any IACUC or permission to use animals for research? Please mention such information at the beginning of the material and methods
Align words (in the ingredients column) in table 1 to the left to make it more organized
Table 2, 3 try to be more organized in the way information is shown
Author Response
In the simple summary and rest of the paper please change the word “killing” by a more professional word such as “disposed, or euthanized…”
This suggested change has been made
Do you have any IACUC or permission to use animals for research? Please mention such information at the beginning of the material and methods
The permission code was added to materials and methods
Align words (in the ingredients column) in table 1 to the left to make it more organized
Table 1 has been formatted to the left to make it more organized
Table 2, 3 try to be more organized in the way information is shown
Tables have been formatted to organize the information

Reviewer 4 Report
The need to evaluate the role for Moringa in specific indigenous breeds because of the strong differences in genetics. The welfare considerations for males of egg laying breeds are also relevant. The poorer FCR is an issue, but the improved breast yield could help to compensate for this.
I have a concern about reporting values to two decimal places. The values should not be reported to an accuracy level greater than can be achieved with the instruments used. For example, reporting LW to 0.01g requires scales measuring birds to this degree of accuracy.
I have suggested acceptance after minor revision with most changes being grammatical.
Author Response
-
The need to evaluate the role for Moringa in specific indigenous breeds because of the strong differences in genetics. The welfare considerations for males of egg laying breeds are also relevant. The poorer FCR is an issue, but the improved breast yield could help to compensate for this.
I have a concern about reporting values to two decimal places. The values should not be reported to an accuracy level greater than can be achieved with the instruments used. For example, reporting LW to 0.01g requires scales measuring birds to this degree of accuracy.
I have suggested acceptance after minor revision with most changes being grammatical.
Instead of 2 decimals, only one decimal place is being reported in the tables of results (except for P-values).

Round 2
Reviewer 1 Report
Your efforts to support your research are appreciated. Unfortunately, the responses provided do not fix the flaws present in the manuscript.
Comment 1: To support the differences in CP are not relevant you are using 3 arguments:
a) That the differences against the control (T1) are numerically small. In this regard, your data have been analyzed as a multiple comparison; therefore, not only the differences T2 and T3 versus T1 matter, but all among the three treatments. The difference in analyzed CP between T2 and T3 is 1.58 percent points; that is a 7.8% difference. That is high enough to confound the responses.
b) That the lack of differences in performance support the fact that the diets had equivalent nutritional value. However, the lack of statistical significance does not prove the absence of effect, and P values higher than the chose threshold do not demonstrate no effect exists. Therefore, that does remove the presence of the different CP values being confounded.
c) That other papers were accepted with similar differences in treatment diets. If the provided references show similar differences (1 percent point in CP among treatments and the response variables are sensitive to changes in CP) they only support the fact that flaws are not always detected.
Comment 2: The only way the metabolizable energy can be the same in the three treatment is if it was a typo mistake. There is no other possible way for the calculated value to be the same, unless moringa metabolizable value was faked in the formulation software.
